# Clinical Predictors of Cognitive Impairment in a Cohort of Patients with Older Age Bipolar Disorder

**DOI:** 10.3390/brainsci15040349

**Published:** 2025-03-27

**Authors:** Camilla Elefante, Giulio Emilio Brancati, Maria Francesca Beatino, Benedetta Francesca Nerli, Giulia D’Alessandro, Chiara Fustini, Daniela Marro, Gabriele Pistolesi, Filippo Baldacci, Roberto Ceravolo, Lorenzo Lattanzi

**Affiliations:** 1Psychiatry Unit, Department of Clinical and Experimental Medicine, University of Pisa, 56126 Pisa, Italy; 2Neurology Unit, Department of Clinical and Experimental Medicine, University of Pisa, 56126 Pisa, Italy

**Keywords:** bipolar disorder, older age bipolar disorder, cognitive impairment, mild cognitive impairment, major neurocognitive disorder, dementia, Alzheimer’s disease, neurodegeneration, vascular leukoencephalopathy, dyslipidemia

## Abstract

**Background**: An increased risk of cognitive decline has been reported in patients with older age bipolar disorder (OABD); however, the underlying factors contributing to this association remain unclear. This cross-sectional study aims to identify the clinical features associated with cognitive impairment in OABD. **Methods**: A total of 152 participants, aged at least 50 years and diagnosed with bipolar disorder (BD) and related disorders in agreement with the Diagnostic and Statistical Manual of Mental Disorders, Fifth Edition, Text Revision criteria, were included in the study and divided into two subgroups based on the presence/absence of cognitive impairment, defined as a diagnosis of Mild Neurocognitive Disorder or Major Neurocognitive Disorder. Univariate comparisons and multivariate logistic regression models were performed to investigate the associations between clinical variables and cognitive impairment. **Results:** Cognitively impaired patients had a higher prevalence of otherwise specified BD/cyclothymic disorder, while BD type 2 was more common in the cognitively unimpaired group. Additionally, the cognitively impaired group had a later onset of major mood episodes (*p* < 0.05), fewer lifetime depressive episodes (*p* = 0.006), a higher prevalence of vascular leukoencephalopathy (*p* = 0.022) and dyslipidemia (*p* = 0.043), a lower prevalence of agoraphobia (*p* = 0.040), worse global functioning (*p* < 0.001), and higher psychopathology severity (*p* < 0.001). Late onset, vascular leukoencephalopathy, and dyslipidemia were all independently associated with cognitive impairment. **Conclusions**: Atypical BD, late onset of mood episodes, and somatic comorbidities like vascular leukoencephalopathy and dyslipidemia are associated with a higher risk of developing cognitive impairment and neurodegenerative disorders in OABD patients.

## 1. Introduction

Older age bipolar disorder (OABD) refers to bipolar disorder (BD) occurring in subjects aged ≥50 years [1]. With population aging, the prevalence of OABD is expected to increase, requiring greater attention to its unique challenges [2,3]. One major issue is the increased risk of cognitive decline and neurodegenerative conditions in individuals with OABD [4]. A systematic review and meta-analysis indicate that individuals with BD face nearly three times the risk of developing dementia compared to the general population and a significantly higher risk compared to those with major depressive disorder [5]. A recent large-scale study from the UK Biobank, involving over 486,000 participants aged from 40 to 69 years, revealed that individuals with BD have the highest risk of developing Alzheimer’s disease (AD) among those with psychiatric conditions, with a hazard ratio of 2.37. Although individuals with major depressive disorder also showed an increased risk of AD, their hazard ratio was comparatively lower at 1.63. These findings suggest the involvement of shared genetic pathways and overlapping neurobiological processes, highlighting the unique vulnerability of individuals with BD to neurodegenerative conditions like AD [6]. Given the significant risk of cognitive decline and dementia in individuals with OABD and the recent availability of therapeutic options for the early stages of AD [7], the identification of clinical predictors for dementia progression has even greater clinical implications.

In the last decades, clinical research has distinguished between early-onset (EOBD) and late-onset BD (LOBD), which are thought to arise from partially different pathophysiological mechanisms and exhibit distinct clinical trajectories. EOBD is strongly associated with a family history of mood disorders [8], whereas LOBD is linked to a high prevalence of somatic comorbidities [9] and brain alterations, such as white matter hyperintensities and brain proteinopathies [10,11,12,13]. Traditionally, LOBD has been defined as BD with an onset from age 50 onward. However, to improve the understanding of BD subtypes, the 2015 International Society for Bipolar Disorder Task Force proposed lowering the cutoff to 40 years. This recommendation was based on the assumption that many neurodegenerative processes may start much earlier in life [1]. Among the contributors to the increased risk of cognitive decline and progression to dementia in BD patients, the recurrence and severity of episodes, the number of hospitalizations, and the duration of treatments received have been the most frequently reported [5,14]. Notably, the number of manic/mixed rather than depressive episodes has been identified as the most reliable clinical marker of neuroprogression in BD patients [14]. A case-register study estimated that the risk of developing dementia increases significantly with the number of prior hospitalizations for affective episodes, with an average increase in risk of 6% per episode in BD [15]. A longer duration of illness and long-term exposure to antipsychotic agents have been associated with lower gray matter volume in older adults with BD [16].

Research on clinical predictors of cognitive impairment in OABD is still limited and presents several shortcomings. Most available data on OABD are derived from studies involving mixed-age cohorts, individuals hospitalized during acute episodes, or small samples, and are further hindered by significant methodological variability [1]. In the present study, we aim to identify the factors associated with the development of Mild Cognitive Impairment (MCI) and/or dementia in a cohort of OABD outpatients. In the Diagnostic and Statistical Manual of Mental Disorders, Fifth Edition, Text Revision (DSM-5-TR) [17], the term dementia has been replaced with Major Neurocognitive Disorder, while MCI is referred to as Mild Neurocognitive Disorder. However, since dementia and MCI are still widely used and recognized in both medical literature and clinical practice, we have chosen to retain these terms in this study. We hypothesize that specific clinical and demographic factors contribute to the development of cognitive impairment in OABD, reflecting a complex interplay between psychiatric history, medical comorbidities, and neurobiological mechanisms. This study differs from previous ones by exclusively including outpatients, ensuring a more homogeneous sample and enhancing generalizability to the broader bipolar population. This approach minimizes confounding effects associated with severe illness and hospitalization. Additionally, the use of a standardized age criterion further refines the sample, strengthening this study’s methodological rigor.

## 2. Materials and Methods

### 2.1. Study Participants

Participants were recruited from patients attending the psychogeriatric outpatient clinic of Psychiatry Unit 2 of the University Hospital of Pisa (Italy) between June 2020 and September 2023 based on the following inclusion criteria: (1) age ≥ 50 years; (2) diagnosis of BD type 1, BD type 2, otherwise specified BD (OS-BD), and Cyclothymic Disorder according to the criteria of the DSM-5-TR [17]. Cognitive impairment diagnoses were established only after systematically assessing and excluding potential reversible causes. These were ruled out through a comprehensive clinical evaluation, which involved a detailed medical history, neurological and psychiatric assessments, laboratory tests (including thyroid function, vitamin B12 and folate levels, and metabolic panel), neuroimaging (MRI/CT when appropriate), and a thorough medication review. Patients with brain tumors, prior ischemic or hemorrhagic stroke, and a diagnosis of Huntington’s disease, Parkinson’s disease, or other neurodegenerative Parkinsonisms were excluded from the sample. Additionally, we excluded all individuals whose cognitive impairment occurred before the onset of BD. All subjects provided written, informed consent for data collection, which was analyzed and presented anonymously in an aggregated form. This study was carried out in accordance with the Declaration of Helsinki. The protocol was approved by the Ethical Committee of the University of Pisa (N. 22537_Perugi).

### 2.2. Clinical Assessments

All patients were assessed by psychiatric trainees with at least two years of experience in the field of psychogeriatrics under the supervision of a senior psychiatrist (L.L.). For each patient, the following data were collected at baseline: (1) demographic data (e.g., age, sex, educational history, marital status); (2) family history of psychiatric and neurodegenerative disorders; (3) medical comorbidities, including MCI diagnosed according to the Petersen criteria [18,19]; (4) neuroimaging alterations; (5) current and lifetime psychiatric disorders according to the DSM-5-TR [17]; (6) course features (e.g., illness duration, polarity at BD onset, history of hospitalization, age at onset of psychiatric symptoms, depressive episodes, hypomanic and manic episodes, etc.). The Clinical Global Impression Severity Scale (CGI-S) was used as a proxy for the severity of psychiatric illness, estimated at baseline and at each follow-up visit [20]. The Brief Psychiatric Rating Scale (BPRS) Expanded Version 4.0 and the Global Assessment of Functioning (GAF) were used to rate symptom severity and level of functioning, respectively [21,22]. BPRS subscales were derived according to the model proposed by Velligan and colleagues [23]. In addition, the Mini-Mental State Examination (MMSE) was also administered according to the usual clinical routine [24].

### 2.3. Statistical Analysis

Descriptive statistical analyses were used to summarize both demographic and clinical characteristics of the sample. Afterwards, patients with and without cognitive impairment (i.e., MCI or dementia) were compared. The Wilcoxon rank sum test was used for continuous variables, after excluding normality using the Shapiro–Wilk test. All continuous variables showed non-normal distributions. Pearson’s chi-squared test or, when appropriate, Fisher’s exact test was used for the comparison of categorical variables. Pairwise, Fisher’s exact test with a false discovery rate correction was used for post hoc comparisons in cases of multinomial variables. Moreover, two logistic regression models, namely, a full multivariate model and a stepwise selected model, were built to estimate the probability of cognitive impairment based on the presence of multiple predictor variables. Logistic regression models were used to predict cognitive impairment among participants, with results presented for both the full multivariate model and the stepwise selected model. The Akaike Information Criterion (AIC) and R^2^ values, as well as the area under the curve (AUC) with 95% confidence interval of the corresponding Receiver Operating Characteristic (ROC) curves were reported for each model to assess the fit and explanatory power. In a second analysis, three groups were compared, namely, patients without cognitive impairment, patients with MCI, and patients with dementia. The Kruskal–Wallis test was used in this case for continuous variables. A statistical significance level of *p* < 0.05 was set for all analyses. All analyses were performed using R Statistical Software Version 4.3.1 (Foundation for Statistical Computing, Vienna, Austria).

## 3. Results

### 3.1. Characteristics of the Sample

The study included 152 elderly patients with BD and related disorders (Table 1). The median age of participants was 76 years, with slightly more females. The Body Mass Index (BMI) had a median of 24.69 among 114 patients out of 152. Education levels varied widely, with a median of 8 years of schooling. Most participants were married, and nearly one-third were widowed.

Most patients (42.1%) had a diagnosis of otherwise specified BD/cyclothymic disorder (OS-BD/CD), followed by BD type 2 (37.5%) and BD type 1 (20.4%). The most common diagnosis among patients with OS-BD was cyclothymic disorder with a history of major depressive episodes (n = 26; 17.1%), followed by major depressive episodes superimposed on a lifelong hyperthymic temperament (n = 12; 7.9%) and short-duration hypomanic episodes accompanied by major depressive episodes (n = 8; 5.3%). Other subcategories included cyclothymic disorder (n = 7; 4.6%), hypomanic episodes without major depressive episodes (n = 7; 4.6%), and manic episodes with insufficient symptoms, with or without mixed features (n = 4; 2.6%).

Cognitive impairment was evident in nearly half of the participants, with 27.0% diagnosed with MCI and 19.1% with dementia. Mixed dementia, including the degenerative components caused by AD and cerebrovascular disease, was the most frequent subtype, followed by frontotemporal dementia and vascular dementia.

### 3.2. Comparisons Between Patients with and Without Cognitive Impairment

The entire sample was divided into two subgroups: patients without cognitive impairment and patients with cognitive impairment (MCI or dementia) (Table 2). Patients with cognitive impairment did not exhibit significant differences in age, gender, BMI, or educational levels compared to those without cognitive impairment. However, marital status revealed significant differences, with those cognitively unimpaired being more frequently unmarried than married, divorced, or widowed compared to the other group (*p_fdr_* < 0.05).

A significant overall difference in BD types was observed between the groups. However, post hoc differences were shown only at the uncorrected level, with a higher prevalence of OS-BD/CD than of BD type 2 in the group with cognitive impairment compared to the unimpaired group.

Among psychiatric comorbidities, agoraphobia was more frequent in the cognitively unimpaired group compared to the cognitively impaired group, whereas no statistically significant differences were observed in lifetime anxiety disorders or substance use disorders. Among somatic comorbidities, vascular leukoencephalopathy and dyslipidemia were more prevalent in the group with cognitive impairment compared to the group without cognitive impairment. Thyroid dysfunction, hypertension, type 2 diabetes mellitus, and obesity did not differ between groups.

No significant differences were observed for first-degree family history of psychiatric disorders, mood disorders, BD or related disorders, anxiety disorders, neurodegenerative diseases, dementia, AD, or Parkinson’s disease. The cognitively impaired group showed a trend toward a later age at the onset of psychiatric symptoms. While the difference in the age at onset of mood symptoms was not statistically significant, significant differences were observed for the age at first major mood episode, first depressive episode, and first (hypo)manic episode, with later onset in the cognitively impaired group.

No significant differences were found between the two groups in illness duration, polarity at BD onset, history of psychosis, or suicide attempts. A significant difference was observed in the history of hospitalization between the two groups, with the cognitively impaired group having a higher proportion of individuals with no hospitalizations and a lower proportion of individuals with single hospitalizations and multiple hospitalizations compared to the cognitively unimpaired group. Post-hoc tests revealed that cognitively impaired patients had significantly more frequently none than a single hospitalization compared to cognitively unimpaired group (*p_fdr_* < 0.05).

The comparison of lifetime mood episodes between the two groups revealed a significantly higher number of lifetime depressive episodes in the cognitively unimpaired group compared to the cognitively impaired group, whereas no significant differences were observed in the number of lifetime manic and hypomanic episodes. A significant difference was also observed in the current clinical state between the two groups. The cognitively unimpaired group had a higher prevalence of patients with a current depressive episode compared to the cognitively impaired group, while the cognitively impaired group had a higher prevalence of patients with euthymia and manic episodes (*p_fdr_* < 0.05). The prevalence of patients with hypomanic episodes was comparable between the two groups.

The psychometric assessment also highlighted significant differences, with the cognitively impaired group showing greater symptom severity and poorer functioning. Global functioning, as measured with the GAF, was significantly lower in the cognitively impaired group, and the total BPRS score was significantly higher, reflecting a greater overall psychiatric burden. While no significant difference was observed in the BPRS Depression/Anxiety subscale, the cognitively impaired group showed significantly higher scores in the BPRS Activation subscale, in the BPRS Negative Symptoms/Retardation subscale, and in the BPRS Psychosis subscale compared to the cognitively unimpaired group. The severity of psychiatric illness, assessed using the CGI-S scale, showed a trend toward higher scores in the cognitively impaired group compared to the cognitively unimpaired group. However, this difference did not reach statistical significance.

Comparisons between cognitively unimpaired patients, patients with MCI, and patients diagnosed with dementia are included in the Appendix A.

### 3.3. Logistic Regression Models Predicting Cognitive Impairment

Clinical variables showing significant differences in univariate comparisons were entered into the multivariate logistic regression model. To avoid multicollinearity, we excluded variables such as the age at first depressive episode, the age at first (hypo)manic episode, hospitalization history, lifetime depressive episodes, and lifetime (hypo)manic episodes from the predictive model. Instead, we selected the age at the first major episode as a key variable. Based on the full multivariate model, including BD type 1 and BD type 2 vs. OS-BD/CD, agoraphobia, late onset of the first major mood episode (≥40 years old), vascular leukoencephalopathy, and dyslipidemia as categorical independent variables, only late onset and vascular leukoencephalopathy were significantly associated with cognitive impairment (AUC = 0.865 [0.787–0.943]). After stepwise selection, agoraphobia, late onset, vascular leukoencephalopathy, and dyslipidemia were retained in the model, with only the latter three being significantly associated with cognitive impairment (AUC = 0.907 [0.842–0.971]) (see Table 3).

## 4. Discussion

In our sample of OABD patients, of whom 63% are female—likely due to the greater longevity of women—atypical BD subtypes, late onset of mood episodes (≥40 years), and somatic comorbidities such as vascular leukoencephalopathy and dyslipidemia were found associated with cognitive impairment. Regarding BD subtypes, our results show that a lifetime diagnosis of OS-BD/CD is significantly more prevalent than BD type 2 among patients with cognitive impairment as compared to the cognitively unimpaired group. In elderly patients presenting bipolar spectrum symptoms, including labile, mixed manic-depressive states, or atypical depressive episodes, but lacking a clear history of canonical BD, these atypical patterns may serve as indicators of significant underlying neurodegenerative processes [25]. Hence, the higher rate of atypical bipolar manifestations belonging to those with OS-BD/CD in the group with cognitive impairment may reflect early behavioral signs of neurodegeneration.

Additionally, our study demonstrates that distinct patterns in the course of BD confer different risks for cognitive decline. Although there is no statistically significant difference in illness duration between the cognitively unimpaired and impaired groups, the former has a disease duration nearly 10 years longer than the latter. A longer disease duration, as well as an earlier onset of mood disorders in the cognitively unimpaired group, could also explain why this group demonstrated a higher frequency of hospitalizations and a higher frequency of depressive episodes. This finding, however, contrasts with the prevailing hypothesis that an increased number of episodes and hospitalizations are associated with accelerated cognitive decline [15,26,27].

In our study, no differences in neurodegenerative or psychiatric family histories were observed between individuals with and without cognitive decline. The lack of difference in the family history of neurodegenerative disorders between the cognitively impaired and cognitively unimpaired groups may be due to the inclusion of different types of cognitive decline within the MCI/dementia group, such as acquired forms with minimal hereditary genetic influence. However, in line with previous research [28,29,30,31,32], when the group with cognitive impairment was further divided into patients with MCI and those with dementia (see Appendix A), it was noteworthy that the dementia group exhibited a higher prevalence of family history of any dementia compared to the rest of the sample. Furthermore, the absence of significant differences in psychiatric family history between the cognitively impaired and cognitively unimpaired groups may be partly explained by the fact that all patients in both groups had a psychiatric diagnosis.

Psychiatric comorbidities showed minimal differences in our sample, with only agoraphobia being more prevalent in cognitively unimpaired patients compared to those with cognitive impairment. However, despite not reaching statistical significance, substance use disorder was nearly twice as common in the cognitively impaired group compared to the cognitively unimpaired group, aligning with the findings of Miguel et al. [33], according to whom substance use disorder associated with BD may exacerbate cognitive decline. In contrast, somatic comorbidities differed between groups. Specifically, cognitively impaired OABD patients exhibited uniquely higher rates of dyslipidemia and a greater burden of vascular leukoencephalopathy compared to their cognitively unimpaired counterparts. Previous reports have also shown somatic comorbidities as risk factors for dementia, such as type 2 diabetes mellitus [34], obesity [35], high cholesterol [36], cerebrovascular pathology [37], and hypertension [38]. A four-year follow-up study identified type 2 diabetes and dyslipidemia as significant contributors to cognitive decline, whereas hypertension did not show a similar impact, partially aligning with our findings [36]. The detrimental effects of metabolic disorders on cognition are believed to stem from the vascular and neurobiological changes they induce, which together may contribute to neurodegeneration [36]. In particular, alterations in lipid levels—especially cholesterol, sphingolipids, and fatty acids—have been linked to AD by promoting amyloid-beta accumulation and tau hyperphosphorylation [39]. In line with our study results, vascular leukoencephalopathy has already been recognized as a risk factor, not only for vascular dementia, but for all types of dementia. The findings of an autopsy-based study, involving 190 elderly individuals and longitudinal aging observations, demonstrated that moderate-to-severe leukoencephalopathy is linked to a substantially higher risk of a clinical diagnosis of dementia, with an OR of 5.4 [37]. Cerebrovascular alterations, in fact, promote amyloid deposition by increasing amyloid-beta production and reducing its vascular clearance, thereby contributing to AD pathology [40]. Addressing these comorbidities could play a critical role in slowing the neurodegenerative process by mitigating their effects on cognitive function.

Based on our findings, the onset of mood episodes, including depressive and (hypo)manic episodes, was significantly delayed in patients with cognitive impairment compared to those without. The onset of BD after the age of 40 was identified as a predictor for the development of cognitive decline, with an OR ranging from 2.55 (1.22–5.32) to 2.59 (1.26–5.34). These findings align with substantial evidence indicating that LOBD may be associated with secondary etiology, predominantly “somatic” factors, which may play a contributory role in the underlying neurodegenerative processes [41,42,43,44]. According to the existing literature, late-onset mood disorders are often associated with brain abnormalities, with vascular changes and neurodegenerative processes potentially leading to late-onset depression [45,46] and (hypo)manic or mixed episodes, as noted by Akiskal et al. [47]. Disruptions in monoaminergic systems, particularly serotonin and norepinephrine pathways, play a critical role in the onset of depression and other neuropsychiatric symptoms during the early phases of neurodegeneration. Neuropathologic changes in AD, which begin long before cognitive symptoms emerge [48], affect regions like the hippocampus and prefrontal cortex, crucial for mood regulation, potentially resulting in mood symptoms as the earliest signs of AD [49]. These results suggest that late-onset psychiatric disorders may represent early manifestations or prodromal symptoms of dementia, emphasizing the importance of timely and accurate differential diagnoses in older adults.

In our study, the severity of psychopathology, as measured with the BPRS, was higher in the cognitively impaired group as compared to the cognitively unimpaired group. It is reasonable to hypothesize that the worsening of psychiatric manifestations, progressing from cognitively unimpaired patients, through those with MCI, to those with dementia, may reflect the cumulative impact of factors contributing to the onset and advancement of neurocognitive disorders. Except for the score on the Depression/Anxiety subscale of the BPRS, which was comparable between the groups, we found higher Negative Symptoms/Retardation, Activation, and Psychosis subscale scores in the cognitively impaired group compared to the cognitively unimpaired group, in line with previous reports [50,51]. The comparison among cognitively unimpaired, MCI, and dementia groups reveals a distinct progressive gradient in the severity of activation, retardation, and psychosis, with these symptoms worsening as cognitive decline advances, whereas levels of depressive and anxiety symptoms remain stable. Similar results were reported by a prospective cohort study [50], which found that the severity of the agitation subsyndrome and apathy increased over the follow-up period. This study also found that more severe dementia was associated with heightened agitation, psychosis, and apathy, but not with more pronounced affective symptoms [50]. Apathy and negative symptoms are common features in multiple neurodegenerative disorders, including AD, frontotemporal dementia, and vascular dementia [38], and are associated with a more diffuse loss of gray and white matter volumes [52]. Since apathy tends to worsen as neurodegeneration advances [53], it is not surprising that it is more pronounced in the cognitively impaired group compared to the cognitively unimpaired group. Moreover, a decline in functioning was observed from cognitively unimpaired individuals to those with dementia, as assessed using the GAF scale. This decline encompasses psychological, social, and occupational domains, highlighting the pervasive impact of neurodegenerative processes on overall functional capacity.

Several limitations of this study should be acknowledged. First, the cross-sectional nature of our study precludes the possibility of establishing causal relationships between clinical variables and cognitive impairment in OABD. Longitudinal studies are needed to confirm the observed associations and investigate causality. Since the study was conducted between June 2020 and September 2023, starting after the first wave of the pandemic, this timeframe may have impacted the type and prevalence of the psychiatric comorbidities observed. Regarding vascular leukoencephalopathy and comorbidities, the neuroimaging procedures performed were not uniform across all patients. Consequently, it was not possible to ensure homogeneity in the neuroimaging findings and, therefore, to precisely quantify the extent of leukoencephalopathy. Some clinical variables, such as the number of lifetime mood episodes—particularly hypomanic episodes—or the age at onset of the mood disorder, were obtained from self-reports or medical records. Relying on retrospective data raises the possibility of recall bias or inaccuracies, which may impact the reliability of the findings. In addition, incorporating specific mood assessment tools, such as the Young Mania Rating Scale [54] and the Hamilton Depression Rating Scale [55], could have provided a more detailed characterization of affective symptoms, allowing for a more precise evaluation of their severity and manifestation in relation to cognitive decline. The clinical setting—being a tertiary psychiatric unit—might not accurately represent all elderly patients with BD. In fact, this sample is likely to represent a subset of individuals at higher risk of cognitive decline. Furthermore, the limited sample size for each dementia subtype necessitated aggregating all subtypes for analysis to examine the relationship between clinical variables and the risk of cognitive decline. This methodological approach may have constrained the ability to determine whether specific BD types, or distinct BD-related features increase susceptibility to particular dementia subtypes. The lack of significant results for some post hoc comparisons may be attributable to low statistical power. Future research should aim to address these limitations by employing longitudinal designs, examining dementia subtypes separately, and incorporating systematic assessments, including neuroimaging and biomarkers, to provide a more nuanced understanding of cognitive impairment in OABD.

## 5. Conclusions

Our investigation highlights that less canonical forms of BD may be more closely linked to cognitive decline compared to BD types 1 and 2. Hence, atypical BD presentations occurring in older age may be considered as prodromal stages of dementia. Additionally, our results indicate that a delayed onset of major mood episodes may be associated with cognitive impairment, supporting the theory of a greater neurodegenerative burden in patients with late-onset mood disorders compared to those with early-onset. As expected, we observed a progressive worsening in general psychopathology severity and a decline in functional capacity across the continuum from cognitively unimpaired patients to those with MCI and, ultimately, to those with dementia. Finally, it is likely that the multifaceted relationship between OABD and neurocognitive disorders encompasses common pathophysiological mechanisms related to the presence of vascular risk factors and the cerebrovascular pathological burden. In line with a precision medicine approach to neuropsychiatric syndromes, identifying specific clinical features in OABD individuals that predict the onset of cognitive impairment could aid in screening patients for second-level investigations and, subsequently, in the selection of those eligible for enrollment in trials targeting disease-modifying treatments.

## Figures and Tables

**Table 1 brainsci-15-00349-t001:** Characteristics of the sample (n = 152). Quantitative variables are presented as median values with 25th and 75th quartiles while qualitative variables are expressed as frequencies and percentages.

Variables	n (%)	M [IQR]
Demographic variables		
Age		76.00 [69.00, 81.00]
Sex (male)	56 (36.8%)	
Schooling years		8.00 [5.00, 13.00]
Marital status		
Unmarried	9 (5.9%)	
Married	84 (55.3%)	
Divorced	16 (10.5%)	
Widowed	43 (28.3%)	
**Mood disorders**		
BD type 1	31 (20.4%)	
BD type 2	57 (37.5%)	
Otherwise specified BD/cyclothymic disorder	64 (42.1%)	
**Psychiatric comorbidities**		
Anxiety disorder	102 (67.1%)	
Panic disorder	70 (46.1%)	
Generalized anxiety disorder	49 (32.2%)	
Separation anxiety disorder	20 (13.2%)	
Agoraphobia	16 (10.5%)	
Social anxiety disorder	2 (1.3%)	
Substance use disorder	23 (15.1%)	
**Cognitive status**		
Cognitively unimpaired	82 (53.9%)	
MCI	41 (27.0%)	
Dementia	29 (19.1%)	
**Type of dementia**		
Alzheimer’s disease	1 (0.7%)	
Frontotemporal dementia	6 (3.9%)	
Vascular dementia	4 (2.6%)	
Mixed dementia	17 (11.2%)	
Undetermined dementia	1 (0.7%)	
**Somatic comorbidities/features**		
Vascular leukoencephalopathy	60 (39.5%)	
Thyroid disease	40 (26.3%)	
Hypertension	78 (51.3%)	
Type 2 diabetes mellitus	25 (16.4%)	
Dyslipidemia	68 (44.7%)	
Obesity	17 (11.2%)	
BMI (n = 114)		24.69 [21.92, 27.22]
**First-degree family history**		
Any psychiatric disorder	107 (70.4%)	
Any mood disorder	82 (53.9%)	
Any bipolar or related disorder	28 (18.4%)	
Any anxiety disorder	37 (24.3%)	
Neurodegenerative diseases	58 (38.2%)	
Any dementia	32 (21.1%)	
Alzheimer’s disease	14 (9.2%)	
Parkinson’s disease	12 (7.9%)	
**Age at onset**		
Psychiatric symptoms		25.00 [18.00, 52.00]
Mood symptoms		35.00 [20.00, 57.25]
First major mood episode (n = 144)		43.50 [27.00, 59.00]
First depressive episode (n = 128)		40.50 [25.75, 55.50]
First (hypo)manic episode (n = 106)		57.00 [35.25, 67.00]
Dementia (n = 29)		75.00 [69.00, 80.00]
MCI (n = 58)		72.50 [66.25, 77.00]
**Illness features**		
Illness duration		36.00 [14.75, 52.00]
Polarity onset (hypo)manic episode (n = 144)	35 (24.3%)	
History of psychosis (n = 142)	38 (26.8%)	
History of suicidal attempts (n = 132)	19 (14.4%)	
History of hospitalization (n = 128)		
Single	24 (18.8%)	
Multiple	37 (28.9%)	
None	67 (52.3%)	
Lifetime depressive episode (n = 146)		3.00 [1.00, 4.00]
Lifetime manic episode (n = 147)		0.00 [0.00, 0.00]
Lifetime hypomanic episode (n = 147)		1.00 [0.00, 2.00]
**Current state**		
Depressive episode	53 (34.9%)	
Hypomanic episode	21 (13.8%)	
Manic episode	7 (4.6%)	
Euthymia	71 (46.7%)	
**Psychometric assessment**		
CGI		3.00 [3.00, 4.00]
GAF		55.00 [40.00, 70.00]
BPRS total score (N = 150)		40.00 [34.00, 47.75]
BPRS Depression/Anxiety		14.00 [9.00, 16.00]
BPRS Activation		13.00 [10.00, 19.00]
BPRS Negative Symptoms/Retardation (n = 151)		7.00 [5.00, 10.00]
BPRS Psychosis		7.00 [6.00, 10.00]

BD: bipolar disorder; BMI: Body Mass Index; BPRS: Brief Psychiatric Rating Scale; CGI: Clinical Global Impression; GAF: Global Assessment of Functioning; IQR: Interquartile Range; M: median value; MCI: Mild Cognitive Impairment; N: frequency.

**Table 2 brainsci-15-00349-t002:** Differences between patients with and without cognitive impairment. Differences are referred to as statistically significant when *p* < 0.05 (shown in bold).

	Cognitively Unimpaired (n = 82)	MCI/Dementia (n = 70)		
	M [IQR]/n (%)	M [IQR]/n (%)	SMD	*p*
**Demographic variables**				
N	82	70		
Age	75.00 [70.25, 79.00]	76.50 [69.00, 81.00]	0.080	0.381
Sex (male)	34 (41.5%)	22 (31.4%)	0.210	0.267
Schooling years (n = 149)	8.00 [5.00, 13.00]	8.00 [5.00, 13.00]	0.119	0.415
Marital status *			0.633	**0.002**
Unmarried	9 (11.0%)	0 (0.0%)		
Married	49 (59.8%)	35 (50.0%)		
Divorced	5 (6.1%)	11 (15.7%)		
Widowed	19 (23.2%)	24 (34.3%)		
**Mood disorders**			0.425	**0.038**
BD type 1	18 (22.0%)	13 (18.6%)		
BD type 2	37 (45.1%)	20 (28.6%)		
OS-BD/CD	27 (32.9%)	37 (52.9%)		
**Psychiatric comorbidities**				
Anxiety disorder	56 (68.3%)	46 (65.7%)	0.055	0.870
Panic disorder	39 (47.6%)	31 (44.3%)	0.066	0.810
Generalized anxiety disorder	26 (31.7%)	23 (32.9%)	0.025	1.000
Separation anxiety disorder	15 (18.3%)	5 (7.1%)	0.339	0.074
Agoraphobia	13 (15.9%)	3 (4.3%)	0.392	**0.040**
Substance use disorder	9 (11.0%)	14 (20.0%)	0.251	0.187
**Somatic comorbidities/features**				
Vascular leukoencephalopathy	25 (30.5%)	35 (50.0%)	0.406	**0.022**
Thyroid disease	22 (26.8%)	18 (25.7%)	0.025	1.000
Hypertension	43 (52.4%)	35 (50.0%)	0.049	0.891
Diabetes mellitus type 2	12 (14.6%)	13 (18.6%)	0.106	0.665
Dyslipidemia	30 (36.6%)	38 (54.3%)	0.361	**0.043**
Obesity	9 (11.0%)	8 (11.4%)	0.014	1.000
BMI (n = 114)	24.77 [22.23, 27.24]	24.38 [21.64, 27.14]	0.101	0.462
**First-degree family history**				
Any psychiatric disorder	58 (70.7%)	49 (70.0%)	0.016	1.000
Any mood disorder	46 (56.1%)	36 (51.4%)	0.094	0.680
Any bipolar or related disorder	13 (15.9%)	15 (21.4%)	0.144	0.500
Any anxiety disorder	19 (23.2%)	18 (25.7%)	0.059	0.861
Neurodegenerative diseases	29 (35.4%)	29 (41.4%)	0.125	0.549
Any dementia	16 (19.5%)	16 (22.9%)	0.082	0.761
Alzheimer’s disease	7 (8.5%)	7 (10.0%)	0.050	0.976
Parkinson’s disease	7 (8.5%)	5 (7.1%)	0.052	0.987
**Age at onset**				
Psychiatric symptoms	23.00 [18.00, 46.00]	30.00 [20.00, 55.00]	0.261	0.066
Mood symptoms	30.00 [20.00, 54.25]	44.50 [23.00, 58.00]	0.223	0.140
First major mood episode (n = 144)	35.00 [23.00, 55.00]	52.00 [31.00, 60.00]	0.492	**0.002**
First depressive episode (n = 128)	35.00 [23.00, 52.00]	50.00 [30.00, 58.00]	0.367	**0.019**
First (hypo)manic episode (n = 106)	53.00 [30.50, 63.75]	58.50 [47.50, 70.50]	0.479	**0.025**
**Illness features**				
Illness duration	38.50 [18.25, 53.00]	29.50 [10.25, 51.00]	0.181	0.332
Polarity onset (hypo)manic episode (n = 144)	17 (21.5%)	18 (27.7%)	0.144	0.507
History of psychosis (n = 142)	18 (24.0%)	20 (29.9%)	0.132	0.551
History of suicidal attempts (n = 132)	12 (16.4%)	7 (11.9%)	0.132	0.621
History of hospitalization (n = 128)			0.528	**0.017**
Single	18 (25.0%)	6 (10.7%)		
Multiple	24 (33.3%)	13 (23.2%)		
None	30 (41.7%)	37 (66.1%)		
Lifetime depressive episodes (n = 146)	3.00 [2.00, 5.00]	2.00 [1.00, 4.00]	0.309	**0.006**
Lifetime manic episodes (n = 147)	0.00 [0.00, 0.00]	0.00 [0.00, 0.00]	0.154	0.909
Lifetime hypomanic episodes (n = 147)	1.00 [0.00, 3.00]	1.00 [0.00, 1.00]	0.091	0.373
**Current state ***			0.562	**0.010**
Depressive episode	37 (45.1%)	16 (22.9%)		
Hypomanic episode	10 (12.2%)	11 (15.7%)		
Manic episode	1 (1.2%)	6 (8.6%)		
Euthymia	34 (41.5%)	37 (52.9%)		
**Psychometric assessment**				
CGI	3.00 [2.00, 4.00]	4.00 [3.00, 4.00]	0.312	0.086
GAF	65.00 [50.00, 80.00]	45.00 [35.00, 60.00]	0.994	<0.001
BPRS total score (n = 150)	37.00 [31.00, 43.75]	44.50 [37.75, 52.00]	0.757	<0.001
BPRS Depression/Anxiety	13.00 [9.25, 16.00]	14.00 [9.00, 16.00]	0.023	0.925
BPRS Activation	12.00 [9.25, 16.00]	15.00 [11.00, 21.75]	0.478	0.002
BPRS Negative Symptoms/Retardation (n = 151)	6.00 [5.00, 7.00]	9.00 [7.00, 12.00]	0.957	<0.001
BPRS Psychosis	6.00 [6.00, 7.75]	8.00 [7.00, 12.00]	0.648	<0.001

BD: bipolar disorder; BMI: Body Mass Index; BPRS: Brief Psychiatric Rating Scale; CGI: Clinical Global Impression; GAF: Global Assessment of Functioning; IQR: Interquartile Range; M: median value; MCI: Mild Cognitive Impairment; N: frequency; OS-BD/CD: otherwise specified bipolar disorder/cyclothymic disorder; SMD: Standardized Mean Difference. * = variables for which Fisher’s test was used instead of the chi-squared test.

**Table 3 brainsci-15-00349-t003:** Logistic regression models predicting cognitive impairment (N = 144).

	A. Full Multivariate Model (AIC = 188.68; R^2^ = 0.12)			B. Stepwise Selected Model (AIC = 187.2; R^2^ = 0.11)		
Variables	Estimate	OR (95% CI)	*p*	Estimate	OR (95% CI)	*p*
(Intercept)	−0.871	0.42 (0.17–1.02)	0.054	−1.293	0.27 (0.13–0.57)	**0.001**
BD type 1 (ref = OS-BD/CD)	−0.565	0.57 (0.21–1.51)	0.257	-	-	-
BD type 2 (ref = OS-BD/CD)	−0.611	0.54 (0.24–1.22)	0.141	-	-	-
Age at first major mood episode > 40 years	0.934	2.55 (1.22–5.32)	**0.013**	0.952	2.59 (1.26–5.34)	**0.010**
Agoraphobia	−1.159	0.31 (0.07–1.32)	0.113	−1.134	0.32 (0.08–1.3)	0.111
Vascular leukoencephalopathy	0.834	2.3 (1.1–4.81)	**0.027**	0.846	2.33 (1.13–4.83)	**0.023**
Dyslipidemia	0.615	1.85 (0.89–3.85)	0.100	0.723	2.06 (1.01–4.21)	**0.048**

AIC: Akaike Information Criterion; BD: bipolar disorder; CI: confidence interval; OR: odds ratio; OS-BD/CD: otherwise specified bipolar disorder/cyclothymic disorder; ref: reference category.

## Data Availability

The data that support the findings of this study are available upon reasonable request from the corresponding author.

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
