# Peer review of "Clinical Predictors of Cognitive Impairment in a Cohort of Patients with Older Age Bipolar Disorder"

_brainsci, 2025, doi:10.3390/brainsci15040349_

Round 1
Reviewer 1 Report
Comments and Suggestions for Authors
The manuscript contains valuable data regarding cognitive impairment in older bipolar patients, in a sample of adequate numbers.However, the authors should revise it taking into account my main concerns and suggestions below.
- Data collection was carried out between June 2020 and September 2023. That is, it began after the first wave of the pandemic, with difficult times afterwards, especially in Italy. This is a limitation and may explain the type and percentage of psychiatric comorbidities observed, which, except for substance use disorders, are anxiety disorders.
- The authors should comment on the peculiarities of samples of aged patients, in terms of survival.
These include the greater presence of women and the low presence of dual disorders (comorbidity with another mental disorder), especially substance use disorders. In this sense, it is of interest that the introduction takes into consideration the recent review of the subject (Miguel et al., 2023, Eur Neuropsychopharmacology, 75:41-58, doi: 10.1016/j.euroneuro.2023.06.006) of the alteration of cognitive functioning prior to the development of dementia in bipolar disorders. Dual disorders are the norm and not the exception in the clinical practice of severe mental disorders such as bipolar disorder.
- All numerical information contained in the tables should be avoided in the results text, limiting it to qualitative description.
In some measurements, it is possible to include them in the text and reduce the content of the information in the tables (for example, demographic variables or psychometric assessment) or consider making two instead of one (tables 1 and 2), the first with general demographic and general clinical data and another specific one with the cognitive data that motivate the study of the manuscript. - It would be extremely interesting to do a sub-analysis of possible differences between sexes, since there is a sufficient sample available, in the characterization of cognitive impairment.
- The current discussion is excessively long and contains a lot of repetition of the results already presented and details of some previous research that can be ignored.
It should focus on theoretical and practical aspects derived from the data obtained.
Author Response
We sincerely appreciate the reviewer’s recognition of the value of the data and the adequacy of the sample size.
Comment 1: Data collection was carried out between June 2020 and September 2023. That is, it began after the first wave of the pandemic, with difficult times afterwards, especially in Italy. This is a limitation and may explain the type and percentage of psychiatric comorbidities observed, which, except for substance use disorders, are anxiety disorders.
The authors acknowledge the reviewer’s observation regarding the timing of data collection. However, as both lifetime and current comorbidities were analyzed together, it is not possible to determine whether the pandemic had an impact on the prevalence of anxiety disorders in this sample.
Comment 2: The authors should comment on the peculiarities of samples of aged patients, in terms of survival. These include the greater presence of women and the low presence of dual disorders (comorbidity with another mental disorder), especially substance use disorders. In this sense, it is of interest that the introduction takes into consideration the recent review of the subject (Miguel et al., 2023, Eur Neuropsychopharmacology, 75:41-58, doi: 10.1016/j.euroneuro.2023.06.006) of the alteration of cognitive functioning prior to the development of dementia in bipolar disorders. Dual disorders are the norm and not the exception in the clinical practice of severe mental disorders such as bipolar disorder.
In our sample, patient age may have influenced gender prevalence. Specifically, 63% of the sample is female, likely due to the greater longevity of women. We have added the sentence: “of whom 63% are female—likely due to the greater longevity of women—" (lines 304-305). Instead, comorbidities were assessed over a lifetime and were not influenced by age at the time of assessment. However, the retrospective nature of the study may have led to inaccuracies in estimating the prevalence of certain conditions, such as substance use disorders. As stated in the limitations, relying on retrospective data raises the possibility of recall bias or inaccuracies, which may impact the reliability of the findings (lines 487-489). Consistent with Miguel et al., as suggested, our sample shows that BD patients with cognitive decline have almost twice the prevalence of substance use disorder compared to those without cognitive decline. To further support this finding, we have added the following sentence: “However, despite not reaching statistical significance, substance use disorder was nearly twice as common in the cognitively impaired group compared to the cognitively unimpaired group, aligning with the findings of Miguel et al. [33], according to whom substance use disorder associated with BD may exacerbate cognitive decline.” (lines 351-355)
Comment 3: All numerical information contained in the tables should be avoided in the results text, limiting it to qualitative description. In some measurements, it is possible to include them in the text and reduce the content of the information in the tables (for example, demographic variables or psychometric assessment) or consider making two instead of one (tables 1 and 2), the first with general demographic and general clinical data and another specific one with the cognitive data that motivate the study of the manuscript.
We acknowledge the reviewer's suggestion and will revise the results section accordingly, ensuring that numerical information from the tables is avoided in the text and limiting it to a qualitative description.
Comment 4: It would be extremely interesting to do a sub-analysis of possible differences between sexes, since there is a sufficient sample available, in the characterization of cognitive impairment.
We appreciate the reviewer's insightful suggestion. Investigating sex differences in the characterization of cognitive impairment is indeed a valuable avenue of research. We may explore this as the primary analysis in a future study.
Comment 5: The current discussion is excessively long and contains a lot of repetition of the results already presented and details of some previous research that can be ignored. It should focus on theoretical and practical aspects derived from the data obtained.
We appreciate the reviewer's feedback. We have revised the discussion to remove redundancies and streamline the content, ensuring a greater focus on the theoretical and practical implications of our findings. Additionally, we have added the following sentence to support our findings with previous literature: "However, despite not reaching statistical significance, substance use disorder was nearly twice as common in the cognitively impaired group compared to the cognitively unimpaired group, aligning with the findings of Miguel et al. [33], according to whom substance use disorder associated with BD may exacerbate cognitive decline." (lines 351-355).
Reviewer 2 Report
Comments and Suggestions for Authors
Dear Editor,
Thank you for granting me reviewing this manuscript, which aims to establish a predictive model for identifying potential risks contributing to the development of cognitive impairment in OABD. Before the manuscript can be considered for publication, several issues need to be addressed:
-
The sentence “Univariate comparisons and multivariate logistic regression models were performed to investigate the associations between clinical variables and cognitive impairment.” appears twice. Please correct this repetition.(Line 20)
-
It is highly recommended to include a flowchart, as is common in studies related to predictive models, to clearly illustrate the study design and analysis process.
-
Multicollinearity is a critical issue in predictive modeling that must be addressed. The authors should eliminate multicollinearity effects prior to feature selection.
-
It appears that the authors combined the results of univariate and multivariate logistic regression analyses and then applied a stepwise selection procedure to obtain the final model. If this is the case, Table 1 suggests that only “Age at first major mood episode > 40 years” and “Agoraphobia” are positive predictors. The authors should clearly state the selection criteria and provide relevant references to support these findings.
-
As this is a predictive model, its clinical utility is paramount. At a minimum, an ROC curve should be provided. Additionally, the clinical value of the model should be highlighted in the discussion section.
-
Although the authors claim that this study is based on previous research, it is unclear why common clinical factors, such as blood indices, were not considered to enhance the model’s clinical applicability. Currently, the model focuses on categorical clinical factors. The authors should clarify this choice or consider integrating more representative clinical data.
Author Response
We sincerely appreciate the reviewer’s thorough evaluation of our manuscript and their valuable feedback. We will carefully address the issues raised to improve the clarity and quality of our work.
Comment 1: The sentence “Univariate comparisons and multivariate logistic regression models were performed to investigate the associations between clinical variables and cognitive impairment.” appears twice. Please correct this repetition.(Line 20)
Thank you for your careful review. We appreciate your attention to detail and will correct this repetition by removing the duplicate sentence.
Comment 2: It is highly recommended to include a flowchart, as is common in studies related to predictive models, to clearly illustrate the study design and analysis process.
Thank you for the suggestion. However, a flowchart is not typically used in studies of this nature, as our study does not involve a stepwise predictive modeling process but rather a cross-sectional analysis of associations. Given this study design, a flowchart would not add substantial clarity or value to the presentation of our methodology.
Comment 3: Multicollinearity is a critical issue in predictive modeling that must be addressed. The authors should eliminate multicollinearity effects prior to feature selection.
We have added the following paragraph in the Results: “Clinical variables showing significant differences in univariate comparisons were entered in the multivariate logistic regression model. To avoid multicollinearity, we excluded variables such as the age at first depressive episode, the age at first (hypo)manic episode, hospitalization history, lifetime depressive episodes, and lifetime (hypo)manic episodes from the predictive model. Instead, we selected the age at the first major episode as a key variable.” (lines 282-287).
Comment 4: It appears that the authors combined the results of univariate and multivariate logistic regression analyses and then applied a stepwise selection procedure to obtain the final model. If this is the case, Table 1 suggests that only “Age at first major mood episode > 40 years” and “Agoraphobia” are positive predictors. The authors should clearly state the selection criteria and provide relevant references to support these findings.
As mentioned in the manuscript: “Logistic regression models were used to predict cognitive impairment among participants, with results presented for both the full multivariate model and the stepwise-selected model.” (lines 154-156).
Table 3 presents both a full model and a stepwise model, where agoraphobia is not significant, while age at first major mood episode > 40 years, leukoencephalopathy, and dyslipidemia remain significant.
The variables included in the model were clinical variables selected based on those that showed significant differences in univariate comparisons (table 2). To clarify this further, we have added the following sentence in the results: “Clinical variables showing significant differences in univariate comparisons were entered in the multivariate logistic regression model.” (lines 282-283).
However, age at onset of first depressive episode, age at onset of first (hypo)manic episode, history of hospitalization, number of depressive episodes, and number of (hypo)manic episodes were excluded to avoid multicollinearity issues (lines 283-287).
Comment 5: As this is a predictive model, its clinical utility is paramount. At a minimum, an ROC curve should be provided. Additionally, the clinical value of the model should be highlighted in the discussion section.
Thank you for the suggestion. In the Methods, we added: "The Akaike Information Criterion (AIC) and R² values, as well as the area under the curve (AUC) with 95% confidence interval of the corresponding Receiver Operating Characteristic (ROC) curves were reported for each model to assess the fit and explanatory power." (lines 156-160). Additionally, in the Results, we added: "Based on the full multivariate model including BD type 1 and BD type 2 vs. OS-BD/CD, agoraphobia, late onset of the first major mood episode (≥ 40 years old), vascular leukoencephalopathy, and dyslipidemia as categorical independent variables, late onset and vascular leukoencephalopathy were significantly associated with cognitive impairment (AUC = 0.865 [0.787-0.943]). After stepwise selection, agoraphobia, late onset, vascular leukoencephalopathy and dyslipidemia were retained in the model, with the latter three being significantly associated with cognitive impairment (AUC = 0.907 [0.842-0.971]) (see Table 3)." (lines 287-295).
Comment 6: Although the authors claim that this study is based on previous research, it is unclear why common clinical factors, such as blood indices, were not considered to enhance the model’s clinical applicability. Currently, the model focuses on categorical clinical factors. The authors should clarify this choice or consider integrating more representative clinical data.
We appreciate the reviewer's comment. This study follows an observational design, meaning that no additional tests beyond those required for routine clinical practice were prescribed. As a result, blood indices and other laboratory parameters were not systematically available for our sample. However, we acknowledge the potential value of incorporating such data and consider it an interesting possibility for a future study on a similar cohort.
Reviewer 3 Report
Comments and Suggestions for Authors
This cross-sectional study has as its objective the identification of clinical features related to MCI and/or dementia in patients diagnosed with old age bipolar disorder, which is an interesting topic, with potentially beneficial clinical and therapeutic implications. Please refer to the observations below:
Lines 17-21- According to DSM 5 TR, there are no diagnoses of dementia, but of major and minor neurocognitive disorders; therefore, to be congruent with the inclusion criteria selected for BD and related disorders, the same nosological framework should be used;
Lines 33-34- Please clarify how the results of a cross-sectional study could have predictive value for the two sets of variables chosen;
Section 2.1- Were substance use disorders (SUDs) considered as an exclusion criterion since they may be a confounding factor for cognitive deterioration? For example, intoxication with substances, neurocognitive disorders induced by SUD, etc. What about neurocognitive disorders due to reversible causes, like hypothyroidism, vitamin B12/folate deficiency, etc.? Were all concomitant treatments allowed, including benzodiazepines or other sedating agents, prior to the administration of cognitive scales?
Section 2.2- Why were only CGI-S, GAF, and BPRS used for the severity of the psychiatric disorder(s) instead of more mood-focused scales and inventories, such as YMRS, HAMD, and MADRS?
Table 1- thyroid disease is mentioned here, but was it considered a potentially reversible cause for neurocognitive disorder due to a general medical condition?
Author Response
We sincerely appreciate the reviewer's recognition of the relevance of our study and its potential clinical and therapeutic implications. We will carefully consider the observations provided and make the necessary revisions to strengthen our work.
Comment 1: Lines 17-21- According to DSM 5 TR, there are no diagnoses of dementia, but of major and minor neurocognitive disorders; therefore, to be congruent with the inclusion criteria selected for BD and related disorders, the same nosological framework should be used;
We appreciate the reviewer's observation. We acknowledge that the DSM-5-TR classifies cognitive impairments as major and mild neurocognitive disorders rather than dementia. However, we used the term dementia because it remains widely used in clinical practice and research when referring to significant cognitive decline affecting daily functioning. Additionally, the term is still commonly employed in epidemiological studies and international guidelines, ensuring better comparability with previous literature.
Comment 2: Lines 33-34- Please clarify how the results of a cross-sectional study could have predictive value for the two sets of variables chosen
The predictive value in this context means that the presence of lifetime atypical BD, late onset of mood episodes, the presence of vascular leukoencephalopathy, and dyslipidemia is associated with a higher likelihood of developing cognitive impairment and neurodegenerative disorders inpatients with older age bipolar disorder. We have revised the sentence as follows: “Atypical BD, late onset of mood episodes, and somatic comorbidities like vascular leukoencephalopathy and dyslipidemia are associated with a higher risk of developing cognitive impairment and neurodegenerative disorders in OABD patients.”
Comment 3: Section 2.1- Were substance use disorders (SUDs) considered as an exclusion criterion since they may be a confounding factor for cognitive deterioration? For example, intoxication with substances, neurocognitive disorders induced by SUD, etc. What about neurocognitive disorders due to reversible causes, like hypothyroidism, vitamin B12/folate deficiency, etc.? Were all concomitant treatments allowed, including benzodiazepines or other sedating agents, prior to the administration of cognitive scales?
Substance use disorders (SUDs) were not excluded, as our analysis aimed to include all potential contributing factors to dementia. However, cognitive impairment diagnoses were made only after first considering and ruling out potential reversible causes, including intoxication with substances, hypothyroidism, and vitamin B12/folate deficiency. This approach ensured that cognitive deficits were not confounded by treatable factors. To clarify this further, we have added the following sentence: “Cognitive impairment diagnoses were made only after first considering and ruling out potential reversible causes.” (lines 110-111). Additionally, it is important to note that the diagnosis of cognitive impairment was not based on screening scales administered during the visit but rather on DSM-5-TR criteria. This methodological choice was made to ensure a more comprehensive and clinically relevant evaluation, avoiding the limitations of brief cognitive screening tools that may be influenced by temporary factors such as medication effects or reversible causes of cognitive deficits.
Comment 4: Section 2.2- Why were only CGI-S, GAF, and BPRS used for the severity of the psychiatric disorder(s) instead of more mood-focused scales and inventories, such as YMRS, HAMD, and MADRS?
While mood-focused scales like YMRS, HAMD, and MADRS offer valuable insight into specific mood symptoms, BPRS not only includes mood-related items but also assesses broader psychopathological dimensions, such as psychotic and negative symptoms. Additionally, GAF and CGI-S provide a comprehensive evaluation of overall psychiatric illness severity and functioning, which is particularly relevant when examining global burden and functional impairment in an older bipolar disorder population.
Comment 5: Table 1- thyroid disease is mentioned here, but was it considered a potentially reversible cause for neurocognitive disorder due to a general medical condition?
We appreciate the reviewer's observation regarding thyroid disease as a potentially reversible cause of neurocognitive disorder. However, the cases in our sample refer to thyroid function alterations with no significant impact on cognitive function. To clarify this, we have added the sentence: “Cognitive impairment diagnoses were made only after first considering and ruling out potential reversible causes.” (lines 110-111). We sincerely thank the reviewer for this valuable suggestion, which has helped improve the clarity and methodological rigor of our work.
Round 2
Reviewer 1 Report
Comments and Suggestions for Authors
Indeed, as I suggested and the authors replied regarding the timing of data collection, “as both lifetime and current comorbidities were analyzed together, it is not possible to determine whether the pandemic had an impact on the prevalence of anxiety disorders in this sample”. It is necessary to mention this in the limitations paragraph of the discussion, which would be elegant and measured given the dates of the evaluations.
Comments on the Quality of English Languageno comments
Author Response
Comment: Indeed, as I suggested and the authors replied regarding the timing of data collection, “as both lifetime and current comorbidities were analyzed together, it is not possible to determine whether the pandemic had an impact on the prevalence of anxiety disorders in this sample”. It is necessary to mention this in the limitations paragraph of the discussion, which would be elegant and measured given the dates of the evaluations.
Answer: Thank you for your insightful comment. We have incorporated the following sentence into the limitations section: "Since the study was conducted between June 2020 and September 2023, starting after the first wave of the pandemic, this timeframe may have influenced the type and prevalence of psychiatric comorbidities observed." (lines 506-508).
Reviewer 2 Report
Comments and Suggestions for Authors
I have no more question at this stage
Author Response
Comment: I have no more question at this stage
Answer: Thank you for your time and review. We appreciate your feedback.
Reviewer 3 Report
Comments and Suggestions for Authors
Thank you for answering to my observations previously raised.
However, I do not agree entirely to the explanation for using outdated terminology like "dementias", since DSM-5 (2013) and DSM 5 TR (2022) are not at all new for clinicians, and many studies have been conducted with the formula "neurocognitive disorders" in the last decade. Also, the fact tat DSM 5TR is mentioned since the "Abstract", it may create confusion to the readers about the mixed nosographic terminology.
Regarding the ruling out of potential confounders and reversible causes for the cognitive decline, it is not clear how were these excluded (lab analysis, interview, etc.)?
In the third place, the use of BPRS, CGI and GAF is a limitation, since other validated scales, more specific for mood symptoms, are available.
Author Response
Comment 1: However, I do not agree entirely to the explanation for using outdated terminology like "dementias", since DSM-5 (2013) and DSM 5 TR (2022) are not at all new for clinicians, and many studies have been conducted with the formula "neurocognitive disorders" in the last decade. Also, the fact that DSM 5TR is mentioned since the "Abstract", it may create confusion to the readers about the mixed nosographic terminology.
Answer 1: Thank you for your comment. In the abstract, we have adopted the suggested terminology, ensuring consistency with current nosographic classifications. Additionally, we have included the following clarification in the text: "In the Diagnostic and Statistical Manual of Mental Disorders, Fifth Edition, Text Revision (DSM-5-TR) [17], the term dementia has been replaced with Major Neurocognitive Disorder, while MCI is referred to as Mild Neurocognitive Disorder. However, since dementia and MCI are still widely used and recognized in both medical literature and clinical practice, we have chosen to retain these terms in this study." (lines 99-104).
Comment 2: Regarding the ruling out of potential confounders and reversible causes for the cognitive decline, it is not clear how were these excluded (lab analysis, interview, etc.)?
Answer 2: Thank you for your comment. We have clarified this point by adding the following sentence: "Cognitive impairment diagnoses were established only after systematically assessing and excluding potential reversible causes. These were ruled out through a comprehensive clinical evaluation, which included a detailed medical history, neurological and psychiatric assessments, laboratory tests (such as thyroid function, vitamin B12 and folate levels, and metabolic panel), neuroimaging (MRI/CT when appropriate), and a thorough medication review." (lines 121-126)
Comment 3: In the third place, the use of BPRS, CGI and GAF is a limitation, since other validated scales, more specific for mood symptoms, are available.
Answer 3: Thank you for your comment. To acknowledge this limitation, we have added the following statement: "In addition, incorporating specific mood assessment tools, such as the Young Mania Rating Scale [54] and the Hamilton Depression Rating Scale [55], could have provided a more detailed characterization of affective symptoms, allowing for a more precise evaluation of their severity and manifestation in relation to cognitive decline." (lines 516-530)
Round 3
Reviewer 3 Report
Comments and Suggestions for Authors
The quality of the manuscript improved